# Serous Borderline Ovarian Tumor Diagnosis, Management and Fertility Preservation in Young Women

**DOI:** 10.3390/jcm10184233

**Published:** 2021-09-18

**Authors:** Marie Carbonnel, Laetitia Layoun, Marine Poulain, Morgan Tourne, Rouba Murtada, Michael Grynberg, Anis Feki, Jean Marc Ayoubi

**Affiliations:** 1Department of Obstetrics Gynecology and Reproductive Medicine, Hospital Foch, Suresnes and University Versailles, 78180 Versailles, France; Laetitia.layoun@hotmail.com (L.L.); marine.poulain@hopital-foch.com (M.P.); r.murtada@hopital-foch.com (R.M.); jm.ayoubi@hopital-foch.com (J.M.A.); 2Department of Pathology, Hospital Foch, Suresnes and University Versailles, 78180 Versailles, France; m.tourne@hopital-foch.com; 3Center of Reproductive Medicine and Biology, Hospital Antoine Béclère, University Paris-Sud, 91400 Orsay, France; michael.grynberg@aphp.fr; 4Department of Obstetrics and Gynecology, HFR Fribourg Hopital Cantonal, 1708 Fribourg, Switzerland; Anis.Feki@h-fr.ch

**Keywords:** borderline ovarian tumor, fertility preservation, conservative surgery, recurrence, diagnosis

## Abstract

Borderline ovarian tumors (BOT) represent about 10 to 20 percent of all epithelial tumors of the ovary. They constitute intermediate lesions between benign ovarian cysts and invasive carcinomas. They often occur in young women of reproductive age, and, albeit with a favorable prognosis, it may recur on the ipsilateral or contralateral ovary. Controversies surround the diagnostic criteria used for their assessment, and the optimal management to minimize their risk of recurrence and/or transformation into malignant carcinoma. Fertility preservation (FP) is considered a priority in the management of these patients, and studies aim at finding the safest and most effective way to help women with BOT history conceive with minimal risk. We present the experience of a single institution in managing three cases of serous BOT in young nulliparous women, followed by a thorough review of the existing literature, highlighting controversies and exploring the possible FP techniques for these women.

## 1. Introduction

Borderline ovarian tumors (BOT) behave as intermediate lesions between benign cystadenomas and invasive carcinomas, making them a separate histologic and clinical entity. They represent 10 to 20 percent of all epithelial tumors of the ovary. They are characterized by an atypical epithelial cell proliferation without stromal invasion [1]. These often occur in young healthy women of reproductive age, with approximately one thirdbeing diagnosed before the age of 40 years [2,3]. Consequently, fertility preservation (FP) has become a major part of the management of these patients [4]. Fortunately, these tumors are often diagnosed at an early stage while still confined to one or both ovaries, considered as stage I BOT according to the International Federation of Gynecology and Obstetrics (FIGO) classification. BOT of all stages combined have favorable 5-year and 10-year survival rates of 95% and 90%, respectively [5]. The management and prognosis of these tumors is dependent on their histologic subtype. Serous and mucinous BOT are the most common types, rarely spreading beyond the ovaries or having a peritoneal involvement, and therefore displaying a better prognosis as compared to invasive carcinoma [6,7]. However, some pathologic features may indicate a worse prognosis in serous BOT, including the micropapillary pattern, bilateral localization, microinvasion, and the presence of extra-ovarian implants. Another challenge lies in the poor preoperative and surgical diagnostic accuracy for BOT, with often more than one surgical intervention required for complete staging, and diagnosis revealed upon histologic examination [6]. Historically, radical surgery was the treatment of choice for BOT; however, there is now a shift towards a more conservative approach given the young age of patients, presentation mostly at an early stage, and low associated mortality rates. However, several studies have reported a higher recurrence rate of the disease with conservative surgical approaches, particularly in the case of unilateral cystectomy. Mortality rates were not significantly increased despite the increased recurrence, especially when patients benefitted from close follow up [7,8,9,10,11,12]. Consequently, there are opposing views in the literature on whether a unilateral cystectomy or a unilateral salpingo-oophorectomy (USO) would be a better option for the conservative management of a stage I unilateral BOT. Controversies also surround the use and safety of FP techniques, and the management of BOT recurrence. Our objective is to address the ongoing controversies on diagnosis, management, and FP of BOT in young women of reproductive age by sharing our experience in a single institution and reviewing the existing literature.

## 2. Materials and Methods

We present the cases of three young women diagnosed with serous BOT, aged between 21 and 32 years. All three patients were managed in our department of Obstetrics and Gynecology at Foch Hospital between the years 2018 and 2020, where they are still followed. The patients’ information and data were retrieved retrospectively from their medical records and analyzed. The first patient is undergoing in vitro fertilization, the second one is waiting for oocyte donation, and FP has not yet been performed for the third. We present a thorough literature review on the available FP options and their outcomes. Written informed consent was obtained from all three patients to review their medical records and publish their imaging and histopathologic results anonymously.

A literature review was conducted by searching databases including Google Scholar, PubMed and Medline using the keywords “borderline ovarian tumor”, “borderline ovarian tumor AND fertility”, “borderline ovarian tumor AND management”, “fertility preservation”, “ovarian tissue preservation”, “oocyte cryopreservation”, “borderline ovarian tumors AND risk factors”, “frozen section”.

## 3. Results

A description of the three cases is available in Table 1.

The first case involves a 32 year-old nulligravid woman with a history of diffuse endometriosis, who presented to our fertility clinic for investigation of primary infertility. On magnetic resonance imaging (MRI), she was found to have a 65 mm left ovarian cyst with endometriotic aspect and tissue component (Figure 1a), diffuse endometriosis, and two endometriomas of 10 and 20 mm on the right ovary. Preoperative cancer antigen 125 (CA-125) was 35.8 U/mL and other tumor markers including CA 19-9, Alpha Fetoprotein (AFP), Carcinoembryonic Antigen (CEA), and CA 15-3 were all within normal range. Laparoscopic exploration revealed a left ovarian cyst filled with chocolate-colored fluid, suggestive of an endometrioid cyst of the left ovary. The cyst was excised without perioperative rupture and the sample was sent for histopathologic analysis. Additionally, two right ovarian endometriomas were punctured and evacuated. Histopathologic analysis revealed that the 40 × 30 × 20 mm^3^ resected left ovarian cyst was a FIGO IA serous BOT with no micropapillary component or tissue invasion. Subsequently, the patient underwent a restaging surgery two months later, including a USO, an omentectomy and multiple peritoneal biopsies. Histopathologic results were negative for abnormal cells. Close surveillance and a trial of in vitro fertilization (IVF) were then recommended to the patient considering the risk of recurrence. At the 6-month post-operative follow up visit, the patient had a non-remarkable findings on physical examination and a CA-125 level of 16 U/mL. Follow-up MRI done 8 months post-operatively revealed a 3 cm cyst with endometriotic aspect and two vegetations measuring 5 and 3 mm on the right ovary. CA-125 was found to be stable at 15 U/mL. A multidisciplinary board meeting was held to discuss subsequent management given the young age and BOT history of the patient. The board’s decision was to perform a right ovarian cystectomy for suspicion of a contralateral recurrence of a BOT, and then to plan a trial of IVF with fertility preservation. Following laparoscopic right ovarian cystectomy, histopathologic results were in favor of a benign luteal cyst of the ovary. Peritoneal cytology and biopsies were negative for any abnormal cells. IVF with FP was subsequently performed; 8 oocytes were retrieved following an antagonist protocol, yielding 6 embryos. Two were frozen and one was transferred with no pregnancy to date. The patient is doing regular clinic follow ups and MRI surveillance every 4 to 6 months.

The second case is that of a 21 year-old nulligravid woman who was seen in clinic for severe chronic pelvic pain. She was found to have on pelvic ultrasound and on MRI (Figure 1b) a left unilocular liquid ovarian cyst of 102 × 100 × 40 mm^3^ with solid papillary projections. Serum CA 125 was 13,607 U/mL, and CA 15-3 was 47.9 U/mL. An 18-FDG positron emission tomography (PET) scan was then done, confirming the presence of the known pelvic cystic mass of with a peripheral tissue component and no other abnormalities. Subsequently, a multidisciplinary board meeting was held to discuss the best management option for this young patient with fertility preservation options. The patient had a left USO with subsequent small laparotomy done to avoid perioperative rupture of the cyst. The specimen was sent to the pathology lab for immediate-section analysis. However, the nature of the cyst could not be determined at that time, so no further action was done. After histopathologic studies and immunostaining, the resected cyst was identified as a serous BOT with no micropapillary component or tissue invasion. Another multidisciplinary board meeting was held; we proposed to the patient a restaging surgery including an omentectomy and peritoneal sampling, followed by the cryopreservation of oocytes. However, the patient refused both propositions for undetermined reasons. She was compliant with regular clinic visits and did a pelvic MRI every 4 to 6 months for surveillance. Four months postoperatively, pelvic MRI was normal and CA-125 was decreased to 15 U/mL. Twelve months postoperatively, a right ovarian cystic mass measuring 15 × 12 × 14 mm^3^ with peripheral vegetations was discovered on MRI. The cyst increased in size, measuring 21 × 19 × 21 mm^3^ with peripheral vegetations seen on 5-month follow-up MRI. Recurrent contralateral BOT was suspected; therefore, a multidisciplinary board meeting was held, recommending a laparoscopic right cystectomy with omentectomy and multiple peritoneal biopsies. Upon laparoscopic exploration, a small infracentimetric nodule was noted over the previous left oophorectomy site, and it was biopsied. A right ovarian cystectomy was then performed with a subsequent small laparotomy to avoid perioperative rupture of the cyst (Figure 2). Additionally, an omentectomy was performed and many peritoneal biopsies were taken and sent to the pathology lab. Histopathologic analysis revealed a 3.5 cm FIGO IC serous BOT with no invasion or micropapillary component. The infracentimetric nodule biopsy corresponded to a non-invasive peritoneal implant associated with the resected BOT. Likewise, the resected omentum and peritoneal biopsies/cytology had no abnormal cells. Following surgery, the patient had a cryopreservation of 23 oocytes. Up to this day, the patient is still being followed up in clinic and doing regular MRI every three to four months.

The third case involves a 27 year old nulligravid previously healthy woman who presented to our clinic for bilateral ovarian cysts found incidentally on ultrasound and MRI (Figure 1c). MRI showed a solid cyst with a fat component measuring 40 × 30 mm^2^ on the right ovary, and a left ovarian cyst occupying the whole ovary, measuring 30 × 10 mm^2^ with a peripheral tissue component. The right ovarian cyst’s characteristics were that of a typical dermoid cyst; however, the left ovarian cyst’s radiologic characteristics raised suspicion for a possible BOT. The patient’s preoperative serum CA-125 level was 62.1 U/mL. A multidisciplinary board meeting was held to discuss the best treatment option for this patient. Upon laparoscopic exploration, the left cyst had benign characteristics. The right cyst had exophytic vegetations that were biopsied and sent to the pathology lab for frozen section analysis. Preliminary results revealed the presence of a serous BOT on the right, requiring a right USO for complete tumour resection. Given the necessity of informing the patient prior to performing a USO, the action on the right ovary was deferred. The patient subsequently had a left ovarian cystectomy with perioperative rupture. Peritoneal biopsies and cytology were also taken and sent to the pathology lab. Surprisingly, histopathologic results revealed the presence of a FIGO IC BOT bilaterally. The left BOT also had focal territories of the micropapillary component measuring <5 mm (Figure 3a). There was no tumour invasion or micro invasion, and peritoneal biopsies were all benign. However, peritoneal cytology was positive for BOT cells. The patient was then scheduled for a restaging surgery seven days later, during which she had a right USO, peritoneal segment resection, multiple peritoneal biopsies, an omentectomy, and an appendectomy. Histopathologic examination confirmed the presence of a 4 cm serous borderline cystadenoma of the right ovary, with endophytic and exophytic vegetations (Figure 3b). Focal areas of micropapillary components with no adjacent tissue invasion were also identified. The 1 cm resected peritoneal segment was identified as a non-invasive implant. The remaining biopsies, appendix and omentum were all negative for BOT cells. The patient had two follow-up MRIs done at 2 and 5 months postoperatively, showing no evidence of recurrence. The patient did not plan to perform any fertility preservation to this day.

## 4. Discussion

### 4.1. Presentation and Diagnosis of BOT

In all three cases, one or more preoperative characteristic raised the suspicion for an ovarian carcinoma versus a BOT. These characteristics include the presence of peripheral vegetations and solid components on MRI (seen in all three cases), as well as a markedly elevated level of CA 125 (case 2). Previously conducted studies have reported that it is not possible to distinguish BOT from early-stage malignant ovarian carcinoma based on imaging and tumor markers alone [6,13,14]. Destructive stromal invasion is no longer necessary for carcinoma diagnosis according to the European Society for Medical Oncology (ESMO) and European Society of Gynaecological oncology (ESGO) [15]. However, BOT must be suspected when there is at least one imaging feature suggestive of malignancy in a predominantly cystic lesion with a regular thin wall, branching or exophytic papillae, presence of ipsilateral normal ovarian tissue, and the absence of significant ascites, peritoneal or omental lesions, or enlarged lymph nodes [13,16]. Ultrasound has a lower sensitivity and specificity than MRI, which remains the gold standard modality to evaluate ovarian masses [17]. Nevertheless the low false positive rate of ultrasound makes it an efficient first line imaging tool [17]. Imaging features along with CA 125 levels, age of the patient and size of the mass are the most commonly used tools to guide surgical management of a suspicious ovarian mass [18]. The use of CA 125 in the context of BOT is controversial. Although it may be elevated in BOT, the CA 125 serum levels should not be used to indicate malignancy nor to determine the nature of the lesion [19]. CA 125 levels have been found to be more increased in serous BOT as compared to other subtypes, in higher FIGO stages and in larger tumor sizes [19]. The latest recommendations by the French National College of Obstetricians and Gynecologists (CNGOF) state that CA 125 level should be used for the follow up of patients with BOT only when baseline CA 125 value is increased preoperatively [19].

Immediate-section sample analysis was inconclusive in case 2. Albeit a valuable tool for the differentiation between benign and malignant ovarian cysts, immediate-section sample analysis has been shown to have less accuracy, sensitivity, and specificity in detecting BOT [19,20,21]. Limiting factors were mostly found to be a larger tumor size (>10 cm) and a mucinous subtype [21].

### 4.2. Management of BOT

The initial management of the three presented cases consisted in a laparoscopic exploration along with conservative surgery given the young age of the patients and the apparent early stage of the disease. A treatment plan was elaborated for all three patients after holding multidisciplinary meetings and discussing the optimal management in order to minimize the disease burden, prevent recurrence, and provide a plan for FP. Unilateral cystectomy was performed for two of the patients who had cysts measuring <10 cm, and a cyst rupture only occurred in case 3. Although laparoscopy has been associated in some studies with a higher risk of cyst rupture [22], it remains the preferred surgical approach for BOT [4]. A large German series from 2013 reported that laparoscopy does not present any disadvantage in terms of relapse rate or overall survival in patients with BOT when compared to laparotomy, whether for initial or restaging surgery [4,23]. It has been well established that conservative surgery sparing the uterus and at least one of the ovaries is the preferred treatment for early-stage BOT, especially in women of young age who wish to retain fertility [24,25,26]. However, it is still controversial whether cystectomy is preferred over salpingooophorectomy. Li et al. [7] reported a significant increase in recurrence rate among the cystectomy group, when compared to the salpingooophorectomy group, in a large meta-analysis that included 2921 patients who underwent fertility sparing surgery for BOT. This may be attributed to residual ovarian disease or potential intraoperative rupture of the cyst [27,28]. The National Comprehensive Cancer Network (NCCN) recommend performing a USO along with a comprehensive surgical staging for BOT lesions [29]. If incomplete surgical staging was performed during initial surgery, the NCCN recommend doing a chest/abdomen/pelvis Computed Tomography (CT) scan with contrast to detect residual disease and/or invasive implants. Moreover, BOT with a serous subtype was shown to be another independent factor for higher recurrence rates and a shorter recurrence interval [7,30]. Another factor leading to a higher risk of recurrence is young age at diagnosis (less than 40 years) [28]. Recurrence rates following fertility sparing surgery range between 10–35%, being significantly higher than following radical surgery [28,31,32]. In 37% of cases, BOT relapses are diagnosed in the first 2 years, and only 10% are diagnosed after 10 years [4,23]. Pregnancy rates in young patients diagnosed with a BOT before the age of 40 were shown to be satisfactory in both the cystectomy and USO groups with no significant difference between the two [32].

The latest French guidelines recommend unilateral or bilateral cystectomy for stage I BOT, along with omentectomy and multiple peritoneal biopsies, when there is suspicion of BOT on imaging or on immediate-section analysis [5]. Fertility-sparing surgery can be safely offered to all stages IA and IC1 BOT and low-grade carcinoma according to the ESMO-ESGO recommendations. This management is safe in patients with conventional low-grade stage IA. It is acceptable for stage IC1 tumors, with half of the recurrences being isolated on the remaining ovary, and they could be rescued by subsequent surgery [15]. An appendectomy is not recommended any more, even in the case of a confirmed mucinous BOT [15]. Early-stage recurrence may also be treated conservatively by another cystectomy, given the favorable prognosis and the young age of the patients [5].

In case 2, a USO was chosen as initial management given the large tumor size and the pathologic ovarian surface. The patient refused subsequent restaging surgery and experienced a contralateral ovarian relapse managed by a cystectomy. Serous BOT has been associated with a relatively higher incidence of peritoneal implants than other subtypes [6]. However, the need for restaging surgery for early-stage serous BOT should be determined case by case. Restaging surgery is recommended in case of micropapillary pattern, or when inspection of the abdominal cavity during initial surgery is considered incomplete [19]. However, in the long run, there is no significant difference in disease-free survival among unstaged, incompletely staged, or completely staged patients [33]. The factors that may predict poor prognosis include a higher FIGO stage, the presence of invasive implants or a micropapillary pattern, residual disease, and stromal microinvasion [26,27].

All three patients are being closely followed up in clinic to this day. The NCCN and CNGOF recommend doing regular follow up visits every 3 to 6 month up to 5 years following surgery, then annual visits if no recurrence is detected [5,29]. In the occurrence of a relapse, the NCCN recommend performing a surgical evaluation along with a debulking surgery if invasive carcinoma is detected [29]. Table 2 displays a comparison between some of the major guidelines in the management and follow-up of BOT.

### 4.3. Fertility Preservation

FP by oocyte cryopreservation was suggested to all three patients given their young age and the feasibility of fertility-sparing surgery. BOT in young women have been associated with an excellent prognosis and low mortality rates [5], suggesting that the main issue in the management of these tumors is deciding an optimal FP plan [34] that would impact the risk of recurrence. Indeed, fertility counselling is becoming a fundamental step in the management of patients with BOT. FP should be discussed and planned by a multi-disciplinary team as soon as a BOT is suspected in a woman who has not completed childbearing [35]. The multi-disciplinary team must include a gynecologic surgeon specialized in oncologic surgery, an oncologist, an experienced radiologist and an expert in reproductive medicine, and should provide a personalized treatment plan according to patient age, ovarian reserve, previous treatments, individual risk of recurrence, and patient preference [35,36]. FP counselling visits should ideally take place before BOT surgery in order to discuss with the chances of spontaneous fertility before and after surgery, fertility preserving options with its limits and risks [36,37]. Systematic review of MRI imaging is useful to determine pre-operatively the degree of possible invasion of one or both ovaries in order to decide with the patient the best option and to avoid restaging surgeries. Geoffron et al. [38] concluded in a recent study that only 25% of FP consultations for malignant and borderline ovarian tumors took place pre-operatively. Moreover, they found that oocyte cryopreservation is an increasingly popular option for women with BOT and ovarian cancer, with only 28.6% of women declining this procedure [38]. Spontaneous pregnancy rates following either cystectomy or USO were reported to be above 80% [36,39,40], creating controversy on the indication and timing of FP in BOT patients. However, spontaneous pregnancy is affected by several factors including underlying infertility prior to BOT treatment, tumor stage, and the number of prior surgeries [36].

Fertility counseling should be preceded by a preoperative assessment of the ovarian reserve by measuring serum anti-Mullerian hormone (AMH) levels and antral follicle count (AFC) by ultrasound, and if timing is adequate, measuring serum follicle-stimulating hormone (FSH) and Estradiol (E2) levels at day 2–5 of the cycle [35]. Ovarian reserve should also be monitored postoperatively and over time, starting 6 months following surgical treatment [35]. Surgery for BOT has been shown to cause or contribute to underlying infertility due to adhesions and alterations of ovarian function [36]. In a large recent retrospective cohort by Delle Marchette et al., each ovarian surgery reduced the chance of spontaneous fertility by 40% [41].

The most commonly used FP technique is oocyte cryopreservation after ovarian stimulation (OS), with a cumulative livebirth rate of 61.9% when 12 oocytes were vitrified in women under 35 years old [42]. Infertility drugs used in OS in women with BOT have been proven to be safe by most available studies [43]. Although some studies found an increased risk of recurrence with the use of infertility drugs [44], the risk was attributed in most cases to underlying infertility, or to the inherent risk of recurrence of the tumour itself [35]. Moreover, Tamoxifen or aromatase inhibitors may be considered during OS to control estrogen levels and therefore reduce the theoretical risk of recurrence [35,36]. However, OS should be avoided in case of recurrent BOT with an invasive ovarian lesion [35].

The use of ovarian tissue cryopreservation and reimplantation is still controversial; more large-scale studies are needed to assess the efficacy and safety of this technique. Transplanting cryopreserved ovarian tissue involves a risk of implanting undetected BOT cells [45]. The main limitation of this procedure is the impossibility of testing cryopreserved ovarian tissues before implantation [45]. Hence, ovarian tissue cryopreservation is not recommended in post-pubertal women to this day, but its use in young women may be discussed if the majority of ovarian tissue is removed with a poor outcome of ovarian stimulation. Another emerging FP technique is the ex vivo collection of immature oocytes from ovarian tissue after surgery, followed by in vitro maturation of the follicles, before vitrification [46]. This technique has been described throughout the case series and case reports, and large-scale studies are lacking to confirm its efficacy.

Even though FP is considered a high priority in women with BOT, there are some instances in which it is contraindicated; this mostly includes recurrent BOT with high-risk features (micropapillary pattern, presence of implants, or microinvasion), or in case of a preexisting low ovarian reserve [36,38]. The decision to perform FP should be taken on a case by case basis, assessing the risks and benefits of existing procedures according to patient characteristics and wishes.

## 5. Conclusions

Borderline ovarian tumours represent a unique clinical entity characterized by a favourable prognosis when diagnosed and managed in a timely manner. These tumours require the involvement of an experienced multi-disciplinary team, in order to elaborate a tailored management plan on a case-by-case basis, offer an optimal treatment, minimize the risk of recurrence. Fertility preservation options should be discussed in young women presenting with a BOT before having completed childbearing. BOT are not always suspected pre-operatively due to their shared characteristics with benign cysts and malignant lesions. This often exposes the patients to more than one surgical procedure for a complete staging. The most common tools used to assess an ovarian lesion suspicious of BOT are MRI or ultrasound features, along with CA-125 levels. The role of intraoperative immediate section analysis in the diagnosis of these tumours is still controversial. Fertility sparing surgery remains the preferred surgical approach for treatment of these tumors, although controversy still exists concerning the use of unilateral cystectomy versus USO for an early-stage BOT in young women. In either case, complete staging including an omentectomy, multiple peritoneal biopsies and peritoneal washings is widely recommended. FP should be initiated as soon as a BOT is suspected, and if possible, with a pre-operative evaluation of the ovarian reserve. The patient should be informed about all the risks pertaining to her fertility following surgery. Oocyte cryopreservation, when feasible, remains the most popular FP technique up to this day. Another less commonly used technique is ovarian tissue cryopreservation, but it has been disregarded by some authors due to the risks entailed. The future of FP in BOT presently revolves around the study of in vitro maturation of immature follicles, avoiding the risks of fertility drugs and the risk of re-introducing malignant cells.

## Figures and Tables

**Figure 1 jcm-10-04233-f001:**
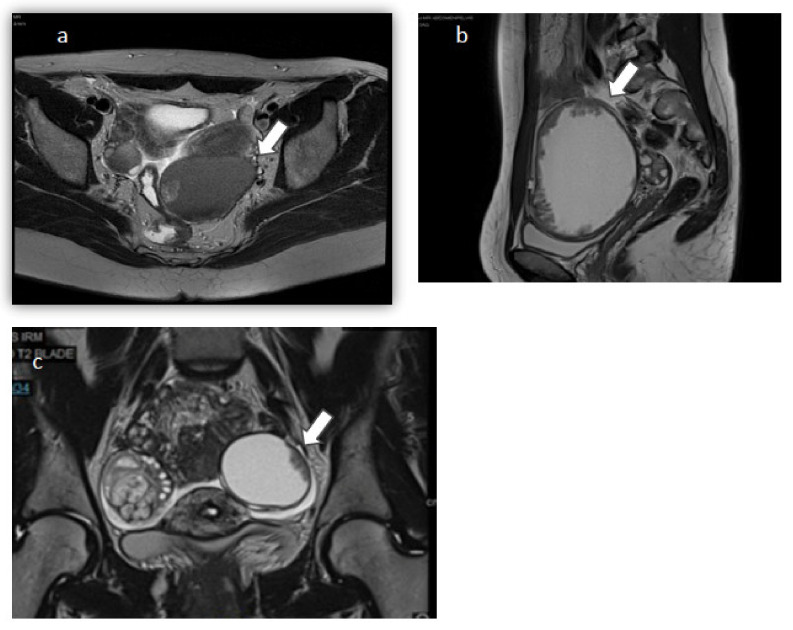
MRI sequences. (**a**). Case 1: Axial T2-weighed MRI sequence. (**b**). Case 2: Sagittal T2-weighed MRI sequence. (**c**). Case 3: Coronary T2-weighed MRI sequence. Arrows pointing at suspicious ovarian cysts.

**Figure 2 jcm-10-04233-f002:**
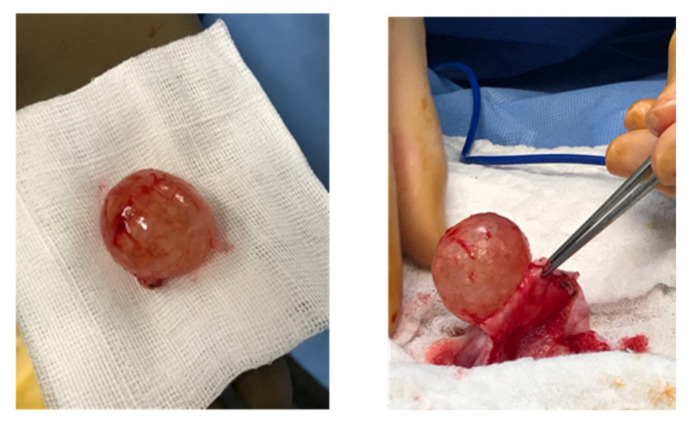
Photograph of a 3 cm recurrence of serous BOT resected after small laparotomy.

**Figure 3 jcm-10-04233-f003:**
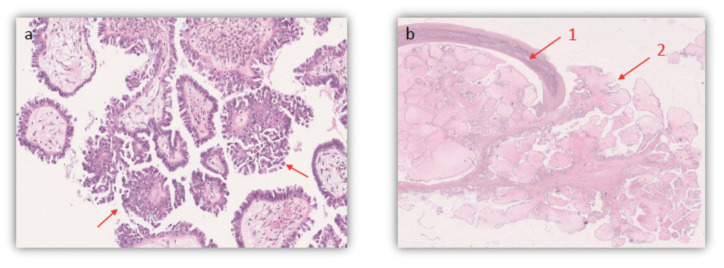
Photomicrography showing histopathologic confirmation of a serous BOT using Hematoxylin and Eosin on FFPE samples (X25.5) in case 3. (**a**). Arrows show Focal areas of micropapillary pattern (**b**). 1. Endophytic and 2. Exophytic vegetations.

**Table 1 jcm-10-04233-t001:** Characteristics and outcomes of three young women with serous BOT.

Case	Case 1	Case 2	Case 3
**Age**	32	21	27
**Parity**	G0P0	G0P0	G0P0
**Past Medical History**	Endometriosis	None	None
**Presentation**	Primary infertility	Recurrent pelvic pain	Asymptomatic
**Ultrasound Findings**	NA	Left ovarian cyst measuring 92 × 109 × 74 mm^3^ suggestive of a hemorrhagic cyst	NA
**MRI Findings**	Left ovary: 65 mm endometriotic cyst with tissue component Right ovary: 2 endometriomas measuring 10 and 20 mm	Left ovary: liquid cyst measuring 100.2 × 100.4 × 70 mm^3^ with solid component and peripheral vegetations	Right ovary: solid cyst with fat component measuring 40 × 30 mm^2^ Left ovary: liquid cyst measuring 30 × 10 mm^2^ with a peripheral tissue component
**Tumor Markers**	CA 125: 35.8 U/mL, CA 15-3: 7.7 U/mL, CA 19-9: 6.8 U/mL, AFP: 2.4 ng/mL, ACE: 0.9 ng/mL	CA 125: 13,607 U/mL CA 15-3: 47.9 U/mL	CA 125: 62.1 U/mL
**Initial Surgery**	Unilateral left cystectomy	Left USO	Left cystectomy and right cyst biopsy
**Surgical Approach**	Laparoscopy	Laparoscopy with subsequent laparotomy	Laparoscopy
**Cyst Rupture**	No	No	Yes during extraction
**Histologic Subtype**	Serous borderline ovarian tumor	Serous borderline ovarian tumor	Bilateral Serous borderline ovarian tumors
**Micropapillary Component**	None	None	<5 mm focal territories of the left lesion
**Invasion or Micro Invasion**	None	None	None, positive cytology
**Immunohistochemistry**	WT1+, P16+ E6H4 +, p53 +	CK7+, p53 5%, Ki67 2%, WT1+	NA
**Restaging Surgery**	Left USO + omentectomy + peritoneal biopsies	None	Right USO + omentectomy + appendectomy + peritoneal biopsies
**Histopathology post Restaging Surgery**	No abnormal cells	NA	4 cm serous borderline cystadenoma of the right ovary with endophytic and exophytic vegetations
**Stage**	FIGO 1a	FIGO 1c	FIGO 1c
**Recurrence**	None	Contralateral recurrence of serous BOT stage FIGO 1a (one year later) with a non-invasive peritoneal implant	None
**Other Findings**	Appearance of a right ovarian cyst with peripheral vegetations on 8 months follow up MRI → Benign corpus luteum cyst after laparoscopic cystectomy and histologic analysis	None	On restaging surgery: Focal areas of micropapillary component + 1 cm noninvasive peritoneal implant
**FP**	Undergoing IVF	Oocyte cryopreservation	No plan yet

G: gestity; P: parity; NA: not applicable; FP: fertility preservation; USO: unilateral salpingo-oophorectomy; BOT: Borderline ovarian tumors.

**Table 2 jcm-10-04233-t002:** Comparison between major guidelines in BOT management.

Recommendations	CNGOF	NCCN	ESMO/ESGO
CA 125 value	Follow-up when increased preoperatively	Follow-up when increased preoperatively	Follow-up when increased preoperatively
Surgical management of serous BOT	Cystectomy (unilateral or bilateral) + peritoneal sampling + omentectomy + peritoneum/appendix inspection	USO/BSO + peritoneal sampling + omentectomy + peritoneum inspection	Cystectomy/USO + peritoneal sampling + omentectomy + peritoneum inspection
Follow-up imaging	Pelvic ultrasound	Pelvic ultrasound	Only performed if clinically indicated
Appendectomy	Performed if mucinous BOT or pathological appendix upon inspection	N/A	Not recommended in BOT
Relapse management	Subsequent cystectomy + peritoneal staging	Surgical exploration + debulking if appropriate	N/A

CNGOF: National college of French obstetricians gynaecologists, NCCN: national comprehensive cancer network, ESMO/ESGO: European society of medical oncology/European society of gynaecological oncology, USO: unilateral salpingoophorectomy, BSO: bilateral salpingoophorectomy, N/A: not applicable.

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
