# Peer review of "Serous Borderline Ovarian Tumor Diagnosis, Management and Fertility Preservation in Young Women"

_jcm, 2021, doi:10.3390/jcm10184233_

Round 1

Reviewer 1 Report

Each case should be described separately and more precisely.
It is worth considering the role of ultrasound in the diagnosis of these tumors.
The discussion is rather a review of selected problems in the diagnosis and treatment of BOT. Recent recommendations of ESMO-ESGO  have not been taken into account.

Author Response

Revisions notes:

Each case should be described separately and more precisely.

Answer: we rewrited completely the results section to describe the 3 cases separately and more precisely.

It is worth considering the role of ultrasound in the diagnosis of these tumors.

We added in the discussion “Ultrasound has a lower sensitivity and specificity than RMI which is the gold standard to evaluate ovarian masses, nevertheless the low false positive rate of ultrasound show that it is a good first line imaging(17)”

The discussion is rather a review of selected problems in the diagnosis and treatment of BOT. Recent recommendations of ESMO-ESGO have not been taken into account.

We included in discussion the recent recommendations of ESMO-ESGO:

 “Destructive stromal invasion is no longer necessary for carcinoma diagnosis according to the European Society for Medical Oncology (ESMO) and European Society of Gynaecological oncology (ESGO) (17)”.

“ Fertility-sparing surgery can be safely offered to all stages IA and IC1 low-grade carcinoma according to the ESMO-ESGO recommendations. This management is safe in patients with conventional low grade stage IA. It is acceptable for stage IC1 tumors, with half of the recurrences being isolated on the remaining ovary and they can be rescued by subsequent surgery (17)”

We modified the sentence “An appendectomy is not recommended any more, even in case of a confirmed mucinous BOT” as appendicectomy is recommended in French guidelines but not in recent ESMO-ESGO ones (17)”.  

Reviewer 2 Report

interesting report about 3 cases with overview of current decision making and tailored treatment plans.

I suggest reviewing this article by a linguist before a second submitting

70

However

88

On magnetic ....  (MRI) , cases 1 and 3 had bilateral ovarian cysts. The cysts on the left side were suspicious: in case 1, aspect of endometriotic cyst with tissue component was identified (figure 1a); in case 2, papillary projection on a liquid unilocular cyst was suspected of malignancy ??? (figure 1b).

97 

Case 3

101

in order to avoid a perioperative rupture 

102

Immediate-section

103

perioperative rupture  .. and a biopsy

104

... a bilateral borderline lesions

107

analysis not studies 

107

in all three cases the diagnosis was FIGO I serous BOT

109
in all three cases restaging surgery was 

136

in her case 

139

have not planned  ???? not clear ????

141

at each step, medical record was discussed at multidisciplinary board 

table 

line presentation case 3 : asymptomatic ?

line surgical approach: laparoscopy with subsequent laparotomy 

148

in all three cases

151 

have reported 

161

invasiveness of the disease 

carcinomatous component ? 

168

for differentiation between 

195

37%

200

The latest French guidelines recommend …

207

And pathologic ovarian surface

209

And experienced a contralateral ovarian relapse managed by a cystectomy

213

On histologic examination

218

On histology

223

That would impact the risk of recurrence

227

Do the authors think that multidisciplinary team should also include gynecologic surgeon specialized oncologic surgery ? experienced radiologist ? what do authors think about systematic review of MRI imaging

232

With its limits and risks

241

Fertility counseling should be preceded by a preoperative

262

Transplanting … involves a risk of …

263

The main limitation of this procedure is the impossibility …

288

Conservative Fertility sparing surgery …

Author Response

Reviewer 2 : Answer

interesting report about 3 cases with overview of current decision making and tailored treatment plans.

I suggest reviewing this article by a linguist before a second submitting

We thank the reviewer for its comments. We made all the asked modifications and An English writer made grammar and language editing before resubmission. You can find here all the modifications performed. As asked by the reviewer 1 we described the 3 cases separately

70

However > modification done

88

On magnetic ....  (MRI) , cases 1 and 3 had bilateral ovarian cysts. The cysts on the left side were suspicious: in case 1, aspect of endometriotic cyst with tissue component was identified (figure 1a); in case 2, papillary projection on a liquid unilocular cyst was suspected of malignancy ??? (figure 1b). > for case 1 :” On magnetic resonance imaging (MRI), she was found to have a 65 mm left ovarian cyst with endometriotic aspect and tissue component (figure 1a), diffuse endometriosis, and two endometriomas of 10 and 20 mm on the right ovary.”, case 2: She was found to have on pelvic ultrasound and on MRI (figure 1b) a left unilocular liquid ovarian cyst of 10.2x10x4 cm with solid papillary projections.case 3 : “MRI showed a solid cyst with fat component measuring 40x30 mm on the right ovary, and a left ovarian cyst occupying the whole ovary, measuring 30x10 mm with a peripheral tissue component. The right ovarian cyst’s characteristics were that of a typical dermoid cyst, however the left ovarian cyst’s radiologic characteristics raised suspicion for a possible BOT”

97 

Case 3> done

101

in order to avoid a perioperative rupture >done

102

Immediate-section>done

103

perioperative rupture  .. and a biopsy >done

104

... a bilateral borderline lesions >done

107

analysis not studies >done

107

in all three cases the diagnosis was FIGO I serous BOT> done

109
in all three cases restaging surgery was >done

136

in her case >done

139

have not planned  ???? not clear ???? > we modified the sentence:” The patient didn’t want to perform any fertility preservation to this day”

141

at each step, medical record was discussed at multidisciplinary board > we added this sentence in each case description

table 

line presentation case 3 : asymptomatic ? > we added it

line surgical approach: laparoscopy with subsequent laparotomy >done

148

in all three cases>done

151 

have reported >done

161

invasiveness of the disease 

carcinomatous component ? > we replaced  it with malignancy

168

for differentiation between >done

195

37%>done

200

The latest French guidelines recommend …>done

207

And pathologic ovarian surface>done

209

And experienced a contralateral ovarian relapse managed by a cystectomy >done

213

On histologic examination >done

218

On histology >done

223

That would impact the risk of recurrence>done

227

Do the authors think that multidisciplinary team should also include gynecologic surgeon specialized oncologic surgery ? experienced radiologist ? what do authors think about systematic review of MRI imaging : >we added “The multi-disciplinary team must include a gynecologic surgeon specialized in oncologic surgery, an oncologist, an experienced radiologist and an expert in reproductive medicine, and should provide…” and “Systematic review of MRI imaging is useful to precise before surgery the degree of possible invasion of one or two ovaries in order to decide with the patient the best option and avoid re-operations”.

232

With its limits and risks >done

241

Fertility counseling should be preceded by a preoperative>done

262

Transplanting … involves a risk of …>done

263

The main limitation of this procedure is the impossibility …>done

288

Conservative Fertility sparing surgery …>done